# Natural Antioxidants from Endemic Leaves in the Elaboration of Processed Meat Products: Current Status

**DOI:** 10.3390/antiox10091396

**Published:** 2021-08-31

**Authors:** Lidiana Velázquez, John Quiñones, Rommy Díaz, Mirian Pateiro, José Manuel Lorenzo, Néstor Sepúlveda

**Affiliations:** 1Centro de Tecnología e Innovación de la Carne (CTI-Carne), Programa de Doctorado en Ciencias Agroalimentarias y Medioambiente, Universidad de La Frontera, Temuco 4780000, Chile; l.velazquez01@ufromail.cl (L.V.); john.quinones@ufrontera.cl (J.Q.); 2Departamento de Ingeniería Química, Facultad de Ingeniería y Ciencias, Universidad de La Frontera, Temuco 4780000, Chile; 3Departamento de Ciencias Básicas, Facultad de Medicina, Universidad de La Frontera, Temuco 4780000, Chile; rommy.diaz@ufrontera.cl; 4Centro Tecnológico de la Carne de Galicia, Avd. Galicia No. 4, Parque Tecnológico de Galicia, San Cibrao das Viñas, 32900 Ourense, Spain; mirianpateiro@ceteca.net; 5Área de Tecnología de los Alimentos, Facultad de Ciencias de Ourense, Universidad de Vigo, 32004 Ourense, Spain; 6Departamento de Producción Agropecuaria, Facultad de Ciencias Agropecuarias y Forestales, Universidad de La Frontera, Temuco 4780000, Chile

**Keywords:** meat, charcuterie, native species, natural preservatives, polyphenols, shelf life

## Abstract

During the last few years, consumers’ demand for animal protein and healthier meat products has increased considerably. This has motivated researchers of the meat industry to create products that present healthier components while maintaining their safety, sensory characteristics, and shelf life. Concerning this, natural plant extracts have gained prominence because they can act as antioxidants and antimicrobials, increasing the stability and shelf life of processed meat products. It has been observed that the leaves of plant species (*Moringa oleifera*, *Bidens pilosa*, *Eugenia uniflora*, *Olea europea*, *Prunus cerasus*, *Ribes nigrum*, etc.) have a higher concentration and variety of polyphenols than other parts of the plants, such as fruits and stems. In Chile, there are two native berries, maqui (*Aristotelia chilensis*) and murtilla (*Ugni molinae Turcz*), that that stand out for their high concentrations of polyphenols. Recently, their polyphenols have been characterized, demonstrating their potential antioxidant and antimicrobial action and their bioactive action at cellular level. However, to date, there is little information on their use in the elaboration of meat products. Therefore, the objective of this review is to compile the most current data on the use of polyphenols from leaves of native plants in the elaboration of meat products and their effect on the oxidation, stability, and organoleptic characteristics during the shelf life of these products.

## 1. Introduction

Processed meat products are highly perishable foods that undergo a series of chemical reactions that, notably, significantly decrease their shelf life [1]. Lipid oxidation is recognized as the main agent responsible for the degradation of meat and processed meat products [2]. However, these biochemical processes together with proteolysis are essential for the synthesis of aromatic compounds (alkanes, aldehydes, alcohols, esters, and carboxylic acid) characteristic of matured meat products [3,4]. One of the most widely used barrier technologies to improve the stability of meat and meat products is the use of synthetic additives, such as butylated hydroxytoluene (BHT), butylated hydroxyanisole (BHA), tertiary butylhydroquinone (TBQH), and propyl gallate (GP), which limit the oxidation and inhibit the microbial growth [5]. Still, some of these compounds, such as BHT, are volatile at high temperatures, and their regular consumption increases the risk of cancer. For this reason, consumers are becoming more conscious and want to be informed about what they buy, avoiding foods with artificial additives while increasing their preferences for functional foods (with clean labels) made from natural, raw materials [6,7].

To cover the demands of new consumers, researchers in the meat industry are focused on creating new formulations with more organic and healthier components [8,9]. In this sense, natural plant extracts are an excellent alternative because of their health benefits and their ability to extend the shelf life of foods [10,11,12]. Foods of plant origin contain compounds, such as flavonoids, vitamins, tannins, organic acids, alkaloids, and phenolic acids, that are effective against oxidative processes and microbial growth in meat [13]. In addition, in some cases, they can act as emulsifiers or provide new sensory characteristics to these products. In this regard, the extracts from the leaves of a wide variety of plants (*Azadirachta indica*, *Bidens pilosa*, *Camellia sinensis*, *Coriandrum sativum*, *Eugenia uniflora*, *Hippophae rhamnoides*, *Ilex paraguariensis*, Juglans regia, *Moringa oleifera*, *Olea europea*, *Origanum vulgare*, *Prunus cerasus*, *Ribes nigrum*, Salvia leriifolia, *Salvia Rosmarinus*, *Satureja hortensis*, *Urtica dioica*, *Vaccinium myrtillus*, among others) have been studied for their in the elaboration of meat formulations, such as frankfurters, ground meat, hamburgers, meatballs, sausages, etc. [14,15,16,17,18].

Recently, the high presence of bioactive compounds in berry fruits has been noted as well as their health benefits, which have been demonstrated through experimental, clinical, and epidemiological evidence [19,20]. Southern Latin America and especially southern Chile is a reservoir of a diversity of native berry species that constitute a potential source of natural compounds for the elaboration of functional meat products. Among these native berries, *Aristotelia chilensis* and *Ugni molinae* Turcz are particularly interesting for their high levels of bioactive compounds. The berries of these plants are consumed as fruits and are also used for the artisanal and industrial elaboration of marmalade, preserves, ice cream, and juice concentrates but also in the pharmaceutical industry as raw material for the elaboration of food supplements [21,22]. Moreover, the international demand for these superfoods has increased considerably, and they are now exported to several countries in Europe, the United States, South Korea, Japan, Australia, and Denmark, among others [23]. This has also increased the by-products generated during the harvesting of these fruits, among which are their leaves. These are not used despite their phytochemical richness, which could make a significant contribution to the production of healthier meat products [20]. On the other hand, there are hardly any studies that include leaf extracts of *Aristotelia chilensis* and *Ugni molinae* for the preservation of processed meat products. Therefore, this review aims to provide a description of the polyphenols present in the leaves of *Aristotelia chilensis* and *Ugni molinae* as well as their in-vitro antioxidant and antimicrobial activity. Furthermore, it compiles the most current data on the use of the other native plants (mentioned above) as potential natural antioxidant and antimicrobial alternatives to synthetic additives in the meat industry, evaluating their effect on the oxidation, stability, and organoleptic characteristics during the shelf life of these products.

## 2. Leaves of Endemic Plants as Ingredients in Processed Meat Products

The natural antioxidants commonly used in the preparation of meat products are extracted from different parts of plants [10,24]. Both the composition and the concentration of these extracts can vary depending on where they have been obtained (fruits, peels, seeds, stems, or leaves). Many studies have characterised, quantified, and compared the polyphenols from different parts of plants. They agree that leaves generally have a greater variety and concentration of these bioactive compounds as well as a higher antioxidant capacity [25,26]. In this regard, extracts from the leaves of a wide variety of fruits, vegetables, and spices have been tested in the form of aqueous and hydroalcoholic extracts or as powders in meat products, such as meatballs, burgers, sausages, and ground meat [8,17,27]. In general, the inclusion of these extracts favours the stability of meat products by prolonging their shelf life (Table 1). Nevertheless, it is very important to evaluate both the composition and the dosage of these extracts in new formulations since, in addition to polyphenols, the leaves can also contain other bioactive compounds (vitamins and nitrogenous species—alkaloids, chlorophyll derivatives, amino acids, and amines) that could affect quality indicators as flavour or act as pro-oxidants [28,29]. This would be the case of alkaloids, which can give meat products a bitter flavour [30]. In this sense, it has been observed that when using concentrations between 200–1000 ppm, a good antioxidant and antimicrobial effect is achieved, enhancing, in some cases, the organoleptic characteristics [17,31].

On the other hand, the conditions under which the polyphenol extraction is performed must also be taken into account. It is known that the extraction conditions (particle size, extraction temperature, solid-solvent ratio, type of solvent, and extraction method) determine the yield and composition of the phytochemicals obtained and their antioxidant and antimicrobial capacities [38,39]. In this respect, it has been reported that some bioactive compounds, such as anthocyanins, phenolic glycosides, and flavonoids, are unstable at high temperatures, leading to their degradation and loss of function [40]. Likewise, the chemical and molecular properties of the solvent are critical factors since, with the appropriate solvent, the selectivity on the target molecules, and thus the extraction efficiency, is increased [26,41]. In particular, it has been observed that carotenoids, tocopherols, polymeric proanthocyanidins, and high-molecular-weight tannins are extracted in organic media due to their lipophilic character [42]. Moreover, some flavonols, such as catechin hydrate and ellagic acid, are best extracted when 50% ethanol is used, while glycosidase flavonoids and gallic acid and its derivatives are more easily extracted with distilled H_2_O [25]. In addition, extraction methods should be evaluated, as some may increase the extraction efficiency compared to conventional methods [43]. 

## 3. Effect of Natural Leaf Extracts on the Quality of Processed Meat Products 

### 3.1. Physicochemical Parameters

The incorporation of natural antioxidants into processed meat products in the form of extracts or powders prevents lipid oxidation and retards microbial growth, resulting in improved stability and longer shelf life [16,26]. However, these compounds can induce variations in other physicochemical, microbiological, and organoleptic parameters that are quality indicators of processed meat products. Among these is pH, which functions as a selective barrier to microbial growth and influences the water holding capacity (WHC) of the processed meat products and, therefore, their texture. Changes in pH during storage can have multiple causes. For example, it has been reported that the decrease in pH can be a consequence of the presence of fatty acids and/or free amino acids in the medium, which are generated as a result of lipolysis or proteolysis that these foods may undergo. Additionally, a significant influence may be the accumulation of acidic secondary metabolites, such as lactic acid, which are formed during microbial growth [42,44]. On the other hand, high pH levels indicate the formation of basic substances, such as ammonia, which are also formed by microbial proteolysis [45]. Several studies demonstrate that there is a maximum pH limit established for meat products. In this regard, a study conducted by Ali et al. [46] showed that beef sausages that had been treated with oregano (*Origanum vulgare*) powder and other natural spices maintained a pH between 6.42–6.51 on the twelfth day of storage, while control sausages (without antioxidants) reached a pH of 6.98 on the sixth day of storage, leading to visible signs of deterioration. In addition, products with pH values higher than 7.4 are discarded due to their poor organoleptic characteristics [16]. Karpińska-Tymoszczyk [35] also observed an increase in pH after three days of storage in duck meatballs treated with rosemary (*Salvia Rosmarinus*) leaf extracts (1%). Conversely, Falowo et al. [32] and Alirezalu et al. [14] perceived a significant decrease in pH in samples of veal meat treated with hydroalcoholic extracts of moringa leaves (*Moringa oleifera*) (0.5 g/kg) and in sausages formulated with 500 ppm of ethanolic extracts of green tea (*Camellia sinensis* L.) and nettle leaves (*Urtica dioica* L.).

Moisture and water activity (a_w_) also have a considerable influence on the stability of processed meat products since these parameters influence the development and metabolic activity of microorganisms, including the production of toxins and the rate of some chemical hydrolytic and enzymatic reactions. The initial percentage of a_w_ of fresh meat used as raw material for the production of processed meat products varies between 0.996–0.997, making it the ideal medium for the growth of pathogenic bacteria. However, these values decrease considerably during the different processes to which the meat is subjected for the elaboration of processed meat products, such as maturation, salting, or dehydration [47]. In fact, these methods are used as a barrier technology to prevent microbial growth. The incorporation of natural extracts will or will not affect the a_w_ depending on the way they are used. It has been observed that when they are used in the form of extracts, they do not significantly influence this parameter [48]. However, when they are included in the form of powders, they can interact with the free water in formulations, causing this parameter to decrease [35]. 

The protein content is another very important indicator since together with lipids, it determines the nutritional value of meat and meat products. Oxidative damage to proteins can lead to losses in nutritional and sensory quality, such as colour or flavour [29,45]. Myoglobin is the protein responsible for the red colour of muscle foods. When this comes in contact with atmospheric oxygen (oxymyoglobin), the colouration of meat becomes bright red, which is characteristic of fresh products and denotes good condition. In contrast, oxidation of myoglobin (metmyoglobin), produced mainly by low-oxygen pressures, leads to a brown colouration, indicating spoilage [29,31,42,49]. Natural antioxidants can slow down these reactions by inhibiting the formation of metmyoglobin. Ouerfelli et al. [8] observed that the application of hydroalcoholic extracts (0.7%) of fresh leaves of *Azadirachta indica* L. reduced metmyoglobin formation by 36.70% and limited the loss of colour in ground meat. On the other hand, polyphenolic extracts can also improve certain parameters that together indicate colour quality in meat and meat products, such as yellowness (b* = +yellow/−blue), redness (a* = +red/−green), and lightness (L* = +light/−dark) [14,34]. Redness (a*) is an extremely important parameter because it can influence the purchase decision of consumers. It is estimated that when the values range between 4.6 and 10.8, the product is perceived as brown [17]. In this regard, a study that measured the influence of pitanga leaves (*Eugenia uniflora*) during the first 11 days of storage of pork patties showed that treated samples had a slower rate of decline in a* values, maintaining a desirable red colour longer [29]. 

An additional method to estimate nutritional deterioration in shelf life studies is to analyse the characteristics of the lipid profile of the formulations, which would be correlated with the formation of hydroperoxides [18]. Both the analysis of fatty acid and cholesterol profile as well as the formation of hydroperoxides in prototypes with phenolic extracts are performed by comparing samples treated with antioxidants and samples without antioxidants (control). This determines the degree to which the antioxidant retards lipid oxidation [17]. In general, it has been observed that lipid degradation occurs during storage for all products independently of the treatment to which they have been subjected (without antioxidants, with natural or synthetic antioxidants). However, samples treated with plant extracts have significantly less degradation [29].

The formation of hydroperoxides (ROOH) is determined by the thiobarbituric acid-reactive substance (TBARS) assay in which malonaldehyde (MDA), a highly reactive compound formed as a consequence of the autolysis of polyunsaturated fatty acids, reacts with thiobarbituric acid, generating a compound with an absorbance maximum at 532 nm [50,51]. It has been established that when hydroperoxides reach the threshold of 2.5 mg MDA/kg sample, a loss of sensory quality and perception of oxidation can be detected by consumers [52]. A more restrictive level of deterioration (0.6 mg MDA/kg) of the rancid flavour in meat products were established by Georgantelis et al. [53]. The results obtained by Nowak et al. [17] and Vargas-Sánchez et al. [51] showed that the action of polyphenolic compounds from plant leaves delayed lipid oxidation in processed meat products. These inhibitions can be up to 80%, inhibiting lipid oxidation even more effectively than synthetic antioxidants, such as BHT and BHA [8,18,34]. Jayawardana et al. [15] observed that green and black tea (*Camellia sinensis* L.) leaf extracts are able to inhibit the formation of hydroperoxides by 67.4% and 65.2%, respectively, while that of the synthetic antioxidant BHT was 49.7%.

### 3.2. Microbiological Changes

Due to their high-protein nature, meat products are classified as highly perishable foods. Spoilage microorganisms (*Pseudomonas*, *Lactobacillus*, and *Enterococcus*) are responsible for the deterioration of meat products, and pathogenic bacteria (*Salmonella enterica*, *Salmonella typhimurium*, *Staphylococcus aureus*, *Campylobacter jejuni*, *Escherichia coli,* and *Listeria monocytogenes*) can put food safety at risk [54,55]. In order to provide safe food to consumers and to avoid microbial spoilage that can cause dramatic economic losses, the meat industry employs different barrier technologies to preserve these foods. Some of these technologies are the use of synthetic preservatives, curing, high pressures, modified atmosphere, ozone, pulsed electric fields, edible coatings or films, and lactic acid bacteria (LAB), among many others [56,57,58,59].

Natural extracts can also be incorporated as a potential source of bactericidal or bacteriostatic agents that can substitute other widely used technologies. Among the commonly used spices with powerful antimicrobial effect are the extracts of cilantro leaves (*Coriandrum sativum*), which showed an inhibitory effect on the growth of aerobic bacteria when they were applied to turkey meatballs (500 ppm). They inhibited the growth of gram-negative microorganisms, such as *Salmonella choleraesuis* spp., *Campylobacter jejuni* and *Escherichia coli*, probably due to contents of linalool in their essential oils [27,60]. Rosemary leaves (*Salvia rosmarinus*), applied at 1000 ppm in turkey meatballs, limited the growth of psychotropic microorganisms, coliforms, and *Clostridium* [35]. Ali et al. [46] observed that tea leaf extracts (*Camellia sinensis* L.) also significantly reduced mesophilic and psychrotrophic bacteria counts when they were incorporated in chicken sausages (500 ppm). A similar behaviour was observed by Nowak et al. [17] in pork sausages, where *Prunus cerasus* L. and *Ribes nigrum* L. berry leaf extracts prevented the bacterial spoilage of mesophilic, psychrotrophic, LAB, and *Brochothrix thermosphacta* during 14 days of storage. According to Lorenzo et al. [29] and Rocchetti et al. [61], *Eugenia uniflora* leaf extracts (250 ppm) showed an inhibitory effect on *Bacillus cereus*, *Escherichia coli*, *Pseudomonas aeruginosa,* and *Staphylococcus aureus* when were applied to pork burgers. However, the most prominent effect was observed on *Salmonella* spp. In ground beef, *Moringa oleifera* leaf extracts had an inhibitory activity against *Escherichia coli*, *Shigella flexinerii*, *Staphylococcus aureus*, *Staphylococcus epidermidis*, *Enterococcus faecalis*, and *Pseudomonas aeruginosa*, while the extracts of cadillo leaves (*Bidens pilosa*) effectively inhibited the growth of *Staphylococcus aureus*, *Staphylococcus epidermidis*, *Bacillus cereus*, *Enterococcus fecali*, *Escherichia coli,* and *Shigella flexinerii* [62]. The phenolic compounds (gallic acid, caftaric acid, catechin, chlorogenic acid, epicatechin, and caffeic acid) present in the extracts obtained from *Bidens pilosa* are closely related to their antimicrobial effects. This is also reflected in the comparative studies carried out between the antimicrobial effects of n-hexane, ethyl acetate, and methanol extracts and reference drugs [63]. 

The high activity of the aforementioned extracts is related to their polyphenol contents. These bioactive compounds have the ability to interact with cell membrane proteins, causing a deformation in the structure and functionality of bacterial cell membranes. In addition, they affect electron transport and metabolic pathways for protein and nucleic acid synthesis [64]. The mechanisms of action of polyphenols on bacteria are illustrated in Figure 1. In the study mentioned above [62], *Bacillus cereus* and *Serratia marcescens* were resistant to the natural extracts of *Moringa oliofera* and *Pseudomonas aeruginosa* and *Serratia marcescens* to *Bidens pilosa*. According to the authors, the resistance of these strains could be related to the structure of the cell wall that restricts the penetration of the plant extracts. In this sense, it has been observed that gram-negative bacteria are more resistant to the attack of these compounds since their membrane is structurally more rigid [65]. It is also important to note that, like the antioxidant activity, the antimicrobial capacity of plant species can vary and is dependent on multiple factors, such as the type of plant, solvent used, extraction methods, concentration of extracts, and bacterial strain evaluated [43]. 

### 3.3. Organoleptic Attributes

In meat and processed meat products, the most visible characteristics that influence consumers’ purchasing decisions are usually those most affected by oxidative deterioration. Oxidation of myoglobin and lipids, especially monounsaturated and polyunsaturated (which are the most exposed to oxidation due to their chemical nature), promote changes in colour and the appearance of volatile compounds (aldehydes, ketones, and alcohols), providing unpleasant odours and flavours to these products [2,42]. Numerous studies support the favourable effects of natural extracts on sensory quality attributes in processed meat products. General acceptability studies are carried out according to the criteria of consumers who identify the quality of prototypes using the parameters that are perceived through the senses when food is consumed (colour, flavour, odour, and texture) [9]. At the same time, there are very precise instrumental techniques that measure the formation of secondary metabolites from lipid oxidation through the formation of volatile compounds [66]. Although there are many lipid-derived products that contribute to the appearance of unpleasant odours in these products (aldehydes, alcohols, esters, furans, ketones, lactones, etc.), hexanal formation is considered the main indicator of lipid oxidation, determining changes in flavour and odour [2]. In this regard, the analysis of volatile compounds derived from lipid oxidation in lamb sausages with *Curcuma longa* L. extracts (250, 500, and 750 ppm) displayed the formation of aldehydes (hexanal, heptanal, octanal, 2-octenal) and alcohols (butanol, 1-pentanol, 1-hexanol, 1-often-3-ol). Although these compounds were identified in all treatments (control, samples with sodium erythorbate, and samples with turmeric extracts), samples treated with turmeric extracts presented lower hexanal concentration at the beginning and at the end of storage. This finding was closely related to the sensory study conducted by consumers since the sausages with 250 and 750 ppm of turmeric extracts maintained odour and good quality up to 12 days of storage. In contrast, the treatments without antioxidants and with sodium erythorbate were accepted only up to day 6 of storage [9]. These results are in agreement with those observed in beef pâté treated with natural extracts of bramble (*Caesalpinia decapetala*) leaves (500 ppm). The researchers pointed out the correlation between hexanal formation and TBARS values as a way to evaluate the development of lipid oxidation and the role of natural extracts in stabilizing oxidation reactions during storage [42]. Beal et al. [33] observed that sensory quality attributes (texture, colour, and overall acceptability) were not affected by the inclusion of bioactive extracts of mate leaves (*Ilex paraguariensis* St. Hil) in Italian-type sausage. Moreover, the best overall acceptability was observed after 60 days of storage. A similar effect was found by Nowak et al. [17] in sausages treated with cherry and currant leaf extracts. The authors observed that the extracts had no negative effects on sensory attributes, but sensory stability was improved compared to the control during 28 days of storage. In addition, neem (*Azadirachta indica*) leaf extracts (700 ppm) had a positive effect on the colour, aroma, acidity, and overall consumer acceptability of beef pâté [8].

## 4. *Aristotelia chilensis* and *Ugni molinae* Leaves as Source of Natural Antioxidants

### 4.1. Ugni molinae Turcz

*Ugni molinae* Turcz, commonly called “murtilla”, “mutilla”, or “murta”, is a wild perennial shrub of the *Myrtaceae* family endemic to southern Chile. This evergreen shrub originates in the mountains of southern Chile and typically grows near the coastal and pre-Andean mountains between the regions of Maule and Los Lagos. *Ugni molinae* produces a globular red berry that is commonly consumed as a fresh fruit. Moreover, the infusions of its leaves are highly valued in Chilean indigenous folk medicine for the treatment of diarrhoea, dysentery, and urinary tract pain [19]. The mechanisms behind this phenomenon were unknown, but recent research has described them. Shene et al. [67] evaluated the bacterial growth in human faecal samples containing aqueous extracts of *Ugni molinae* Turcz leaves. The results showed that the aqueous extracts had prebiotic effects, favouring the growth of bacteria, such as *Lactobacillus* and Bifidobacteria, which demonstrates that these extracts can be used for therapeutic purposes. On the other hand, several studies also demonstrated the clinical properties of *Ugni molinae* Turcz leaves, concluding that leaf extracts increase plasma antioxidant capacity due to the high amounts of polyphenols identified [68]. In addition, they have an inhibitory effect on enzymes involved in glycaemic control, such as α-amylase and α-glucosidase [69]. 

Although the main commercial use of *Ugni molinae* Turcz is focused on the sale of marmalades, juice, or liquor [70,71], they have also been incorporated into cosmetics and pharmaceutical supplements. In addition, the food industry is studying the use of its extracts for the elaboration of edible films [72].

#### 4.1.1. Bioactive Compound Profile of *Ugni molinae* Turcz Leaves

High-Performance Liquid Chromatography-Mass Spectrometry (HPLC-MS) analysis revealed the presence of a wide diversity of bioactive compounds in the leaves of *Ugni molinae* Turcz (Figure 2a). 

Among the compounds identified are kaempferol, which is a flavonol biosynthesized from naringenin (flavone) in the presence of the enzyme flavone 3-hydroxylase [77]. In addition, ninety-two different compounds have been reported in ethanolic extracts of *Ugni molinae* Turcz leaves, with the main ones as follows: gallic acid, myricetin, myricetin glucoside, myricetin hexoside, myricetin rhamnoside, myricetin dirhamnoside, quercetin, quercetin hexoside, quercetin rhamnoside, quercetin dirhamnoside, quercetin glucoside, epicatechin, and kaempferol glucoside. In the aqueous extracts, in addition to the above-mentioned compounds, gallic acid derivative, myricetin xyloside, and quercetin xyloside were found, although in a lower concentration. Additionally, *Ugni molinae* Turcz leaves are rich in triterpene acids, such as apostolic acid, corosolic acid, asiatic acid, betulinic acid, oleanolic acid, ursolic acid, medacassic acid, ellagic acid, chlorogenic acid, gallic acid, and caffeic acid. In summary, the polyphenols present in *Ugni molinae* Turcz leaves can be divided into flavonol glycosides, non-glycoside flavanols, and triterpenoids [68,71,78].

Several researchers have reported that polyphenolic concentration and composition are variable depending on the different fractions. In this regard, Shene et al. [67] reported that the extraction with 50/50 (*v*/*v*) ethanol/water allows a higher release of polyphenols compared to the extraction with water (8406 and 6876 mg GAE/L, respectively). According to the authors, this is due to the high capacity of low-polarity solvents to extract specific phenolic compounds. Studies conducted by Rubilar et al. [79] and López de Dicastillo et al. [25] on *Ugni molinae* Turcz leaf extracts showed that a higher yield of flavonoid glycosides, such as quercetin 3-O-glucoside, quercetin 3-O-rhamnoside, and gallic acid and its derivatives, are obtained in water, while flavonols, such as catechin hydrate and ellagic acid, are extracted exclusively by alcohols. In addition, Arancibia-Radich et al. [68] observed considerably higher total polyphenols (PT) levels in ethanolic extracts obtained from leaves of 10 different genotypes of *Ugni molinae* than those extracted with ethyl acetate (157.6–225.8 and 36.1–92.3 mg GAE/g extract, respectively). In crude extracts of *Ugni molinae* Turcz leaves, the PT concentration found was higher than those obtained in other plant matrices that have already been used for the elaboration of meat products (634.6 mg GAE/g of lyophilized extract vs. 229.38, 3.17 and 2.17 mg GAE/g for *Ugni molinae* Turcz, pitanga, cherry, and currant leaves, respectively) [17,29,80].

#### 4.1.2. Antioxidant and Antimicrobial Capacity of *Ugni molinae* Turcz Leaves

In this review, the use of leaves is highlighted because several authors have reported that they present higher PT levels and therefore higher antioxidant and antimicrobial capacity. The leaves of *Ugni molinae* Turcz have PT concentrations four-times higher than its fruits (6.81 vs. 2.32 mg/g for leaves and fruit, respectively). Similarly, the antioxidant capacities of the plant suggest the same tendencies (361 vs. 110 mg Trolox/g for leaves and fruit, respectively) [25]. Indeed, a Pearson correlation analysis confirmed a significant positive correlation between PT concentration and antioxidant activity, with a higher correlation for the Trolox equivalent antioxidant capacity (TEAC) and 2,2-diphenyl-1-picrylhydrazyl radical scavenging activity (DPPH) [25]. In addition, Rubilar et al. [69] found that the leaves had higher PT concentration than fruits and stems (32.5, 15.8, and 10.1 mg GAE/g for leaves, stems, and fruits, respectively), which results in a higher antioxidant activity. In this regard, the concentration of crude extract of *Ugni molinae* Turcz leaves required to inhibit 50% of DPPH was two- and four-times lower than that needed by stems and fruits, respectively.

About antimicrobial activity, several authors reported that Ugni molinae Turcz leaf extracts have antimicrobial activity on Escherichia coli, Listeria monocytogenes, Bacillus subtilis, Micrococcus luteus, Staphylococcus aureus, Staphylococcus epidermidis, Pseudomonas aeruginosa, Enterobacter aerogenes, and Candida albicans [25,72,81]. Moreover, phenolic extracts of Ugni molinae leaves incorporated into edible films inhibited the growth of Listeria innocua up to 60 days after storage. These antimicrobial effects are associated with the presence of quercetin in the leaves of Ugni molinae [72]. In addition, they promote the growth of beneficial bacteria, such as Lactobacillus and Bifidobacteria [67]. However, there is still very little information on the minimum inhibitory concentration (MIC) and minimum bactericidal concentration (MBC) of these extracts to inhibit the growth of pathogenic bacteria. 

### 4.2. Aristotelia chilensis (Mol.) Stuntz

*Aristotelia chilensis* is an evergreen tree belonging to the *Eleocarpaceae* family. The genus *Aristotelia* is represented by five species that are distributed in the South Pacific region (Chile, Argentina, New Zealand, Australia, and Tasmania Island). In Chile, this plant, commonly known as maqui, has a geographical distribution that goes from Limarí (IV Region) to Aysén (XI Region). It grows mainly in humid soils of the central valley, on the mountain slopes, ravines, or forest margins from near sea level up to an altitude of 2500 m. Its fruit is a round, fleshy, and dark violet edible berry [82,83]. Its leaves and stems have been traditionally used by indigenous peoples as phytotherapy to prevent or cure some ailments, such as fever, inflammations, dysentery, and haemorrhages [84]. Recent studies have shown that the antioxidant, antidepressant, anti-inflammatory, and anticarcinogenic capacity of this plant are the result of the high biological activity of its compounds at the cellular level [74,85]. Some of these compounds are caffeic acid, ferulic acid, and protopine (isoquinoline alkaloid), which are identified as possibly responsible for the analgesic effect of *Aristotelia chilensis* leaves [84]. Furthermore, Ojeda et al. [86] demonstrated that *Aristotelia chilensis* extracts can reduce the expression of cyclooxygenase 2 (COX-2), preventing the inflammation in colon cancer cells. The alkaloid fraction present in the leaves of this plant (aristothelin and 8-Oxo-9-dihydromakomachin) is considered to have vasodilatory effects on vascular tissues, suggesting that it could be very useful in treating diseases, such as hypertension [87,88]. Furthermore, research in murine models indicated that anthocyanin-rich extracts of *Aristotelia chilensis* act on lipid metabolism by lowering low-density lipoprotein (LDL) levels in the blood and on carbohydrate metabolism due to their anti-diabetic activity [89,90].

The fruit of *Aristotelia chilensis*, besides being consumed, is commonly used in Chile as raw material for the artisanal production of marmalades and preserves. At the industrial level, it is used for the production of natural colourants, ice cream, marmalades, and juices [22,91]. In the pharmaceutical industry, several formulations made from the fruit of *Aristotelia chilensis* have been patented (Delphinol^®^, MaquiBright^®^, among others) [21,92]. Its leaves are used as an unguent, as an infusion to cure throat diseases, intestinal tumours, and to wash out mouth ulcers [82]. The Chilean Ministry of Health’s list of traditional herbal medicines approves the use of *Aristotelia chilensis* leaves to relieve diarrhoea, dysentery, colds, sore throats, inflammation of the tonsils, and mouth ulcers [93]. All these benefits of the *Aristotelia chilensis* plant as a superfood have generated a worldwide demand [94]. Currently, 19 companies are exporting *Aristotelia chilensis* to 9 countries among which South Korea stands out, with more than 80% of the national production (104,000 tonnes) [83]. However, the by-products generated are not utilised yet despite being a rich source of natural compounds that could be used for the production of value-added products based on sustainable technologies.

#### 4.2.1. Bioactive Compound Profile of *Aristotelia chilensis* Leaves

Although the fruits of *A. chilensis* are valued for their high concentrations of polyphenols among which anthocyanins stand out [95], its leaves have a higher concentration and variety of bioactive compounds [69]. In this regard, the leaves of *Aristotelia chilensis* are a source of alkaloids (airstone, aristotelone, aristotelone, aristotelinin, hobartine, makonine, 8-oxo-9 dehydrohobartin, and 8-oxo-9 dehydromakomachin), anthocyanins (pelargonidin (14.45%) and peonidin (0.20%)), flavonols (catechin (21.75%), quercetin (1.3%), isoquercitrin (0.35%), myricetin (2.26%), rutin (1.79%), and kaempferol)), phenolic acids (gallic acid (47.55%) and coumaric acid (6.81%)), and stilbenes (resveratrol (3.55%)) [69,96,97] (Figure 2b). This wide diversity of bioactive compounds provides a high antioxidant capacity to the plant. However, it must be taken into account that both the photochemical composition and the antioxidant capacity will be influenced by biotic and abiotic factors surrounding the plant. A very significant biotic factor is usually the genotype and age of the plant [94,98]. Thus, it has been observed that the capacity of *Aristotelia chilensis* to synthesize anthocyanins varies in young leaves and fully expanded leaves, the latter having a higher concentration of anthocyanins [99]. Abiotic factors are usually influenced by the geographical area and climatic conditions, including solar radiation, humidity, temperature, altitude, water and microbial interactions, soil fertility, and harvest year [96,98]. Considering the above, the climatic conditions under which this plant grows could explain its high concentrations of phytochemicals. Its presence is common in the foothills of the Andes, which are under constant abiotic stress caused by the humidity and cold temperatures typical of this type of climate. This causes a metabolic imbalance that can cause cellular oxidative stress, forcing the plant to produce greater amounts of bioactive compounds [82].

#### 4.2.2. Antioxidant and Antimicrobial Capacity of *Aristotelia chilensis* Leaves

In meat and meat products, lipid degradation, proteolysis, and microbiological contamination lead to rancidity and changes of colour, flavour, and reduction of shelf life [2,100]. These changes occur when these foods are subjected to changing conditions, such as exposure to oxygen, high temperature, and light during their storage as well as the presence of catalytic agents, such as iron [8,50]. In particular, fatty acids are the most susceptible to oxidative degradation and, within this, polyunsaturated acids (PUFA) due to the several double bonds present in their carbon chain are the most susceptible to oxidative degradation [2,101]. In these phases, the presence of natural antioxidants, which prevent or delay the formation of free radicals, is essential. The mechanisms responsible for the antioxidant capacity of these natural extracts are diverse. However, it is well known that the antioxidant richness of natural extracts is proportional to the concentration of bioactive molecules present in them and the interactions between these molecules and the main mechanisms described in food, which involve the transfer of a proton or an electron to a free radical molecule (Figure 3) [25]. 

Natural extracts of *Aristotelia chilensis* leaves show a high polyphenolic concentration and antioxidant activity in all their derived fractions. Moreover, it was also evidenced that the presence of different classes of bioactive compounds varied depending on whether methanol, hexane, dichloromethane, or water was used. In this sense, Rubilar et al. [69] reported that the PT concentration of the crude extract obtained from *Aristotelia chilensis* leaves using 50% (*v*/*v*) ethanol and a solvent-to-solid ratio (5:1) was 69.0 mg GAE/g, while contents of 15.9 and 5.6 mg GAE/g were obtained in the aqueous and organic-aqueous fractions, respectively. Moreover, this study showed that independently of the solvent used for the extraction, leaves had a higher concentration of polyphenols than fruits and stems. Other studies confirmed that when a higher concentration of ethanol was used in the hydroalcoholic mixture (60:40) with a solvent-to-solid ratio (6:1), a higher concentration of PT (40 mM EAG) was found. The same trend was observed in the antioxidant capacity of *Aristotelia chilensis* leaf extracts. DPPH method displayed values of 158.2 and 189.5 mM EAG for extracts at 0.1% and 1%, respectively, while ABTS method showed values of 165.1 and 165.7 mM EAG for extracts at 0.1% and 1%, respectively [97]. Additionally, according to Muñoz et al. [84], the antioxidant activity of *Aristotelia chilensis* leaf extracts against DPPH radical inhibition is higher when water or methanol are used as extraction solvents (12.1 and 9.7 µg/mL, respectively). In the same way, Avello et al. [102] reported that aqueous extracts of *Aristotelia chilensis* leaves had a PT concentration lower than those observed in hydroalcoholic extracts (0.032 vs. 0.040 M EAG, respectively). However, 100% inhibition was observed in DPPH (6.0–0.6 μM EAG) for all extracts.

Likewise, the high antioxidant activity of *Aristotelia chilensis* leaf extracts, expressed as IC_50_ (8.0, 26.9, and 29.4 mg extract/L for crude extracts, aqueous, and organic-aqueous fraction, respectively), was also corroborated [69]. According to the authors, the required concentration of crude *Aristotelia chilensis* leaf extract to inhibit 50% of DPPH was five-times lower than that needed by stem and 50-times lower than that of crude fruit extracts. However, it should be noted that with the use of solvents, it is difficult to obtain a polyphenolic yield of 100% compared to the use of powders. Furthermore, it must be taken into account that methanol is not a useful solvent for biological purposes. Therefore, food studies with *Aristotelia chilensis* leaf powders should be carried out as a way to increase yield, reduce production costs, and offer consumers functional and chemical-free foods. 

Regarding the antimicrobial activity of *Aristotelia chilensis* leaves, it was found that there is little information on their antimicrobial properties. There was only one study carried out by Avello et al. [102] where ethanolic extracts (60%) of *Aristotelia chilensis* leaves showed an inhibitory effect on *Pseudomonas aeruginosa*, *Enterobacter aerogenes*, *Staphylococcus aureus,* and the fungus *Candida albicans*.

## 5. Conclusions

Currently, polyphenols are of great interest to the meat industry because they may have antioxidant and antimicrobial activity that is equal or superior to synthetic antioxidants. Their incorporation into meat products can improve their stability and extend the shelf life of these highly perishable foods. Furthermore, there is sufficient scientific evidence that confirms the association between their regular consumption and the prevention of various chronic diseases, such as inflammation, cancer, diabetes, and Alzheimer’s disease. *Aristotelia chilensis* and *Ugni molinae* Turcz are two endemic plants from Chile with great potential to be used in meat formulations. They distinguish themselves from other fruits and plant species due to their high polyphenolic concentration and in-vitro antioxidant activity. In particular, the leaves of both plants (especially the raw fractions or powders) have higher variety and levels of bioactive compounds, which could be used for the creation of new value-added meat formulations, which demonstrate that the by-products of the agricultural industry can be used as a source of natural antioxidants and antimicrobials. Moreover, many studies have been carried out to characterize the phenolic composition of these plants and their clinical effects. However, little research was found on the antimicrobial potency and, in particular, in MIC and MBC of *Aristotelia chilensis* leaf extracts. In addition, it is necessary to carry out studies in which the extracts of the leaves of *Ugni molinae* Turcz and *Aristotelia chilensis* are incorporated into meat formulations. The evaluation of the effect of these natural extracts on physicochemical, microbiological, and organoleptic parameters will allow to corroborate their great potential as antioxidants and antimicrobials.

## Figures and Tables

**Figure 1 antioxidants-10-01396-f001:**
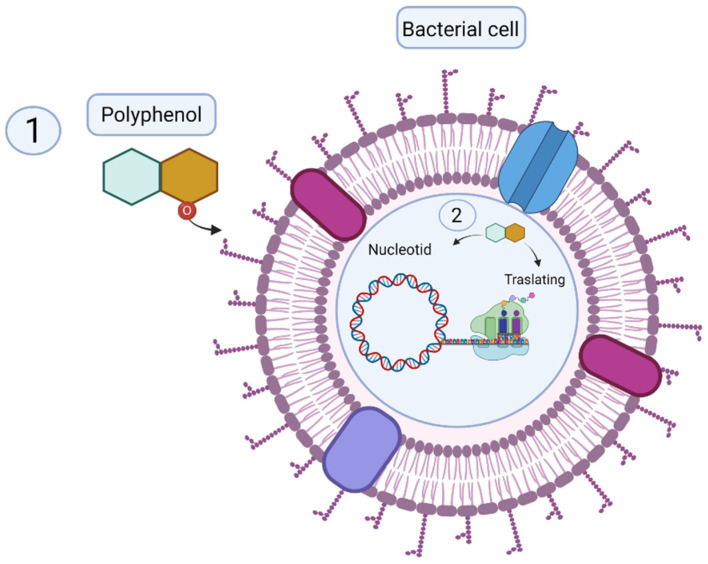
Antimicrobial mechanism of polyphenols. (**1**) The polyphenol molecule reacts with the proteins of the cell membrane of bacteria, causing lysis. (**2**) Inside the cell, it interferes with the synthesis of DNA and proteins, causing the death of the cell (created with BioRender.com, 3 July 2021).

**Figure 2 antioxidants-10-01396-f002:**
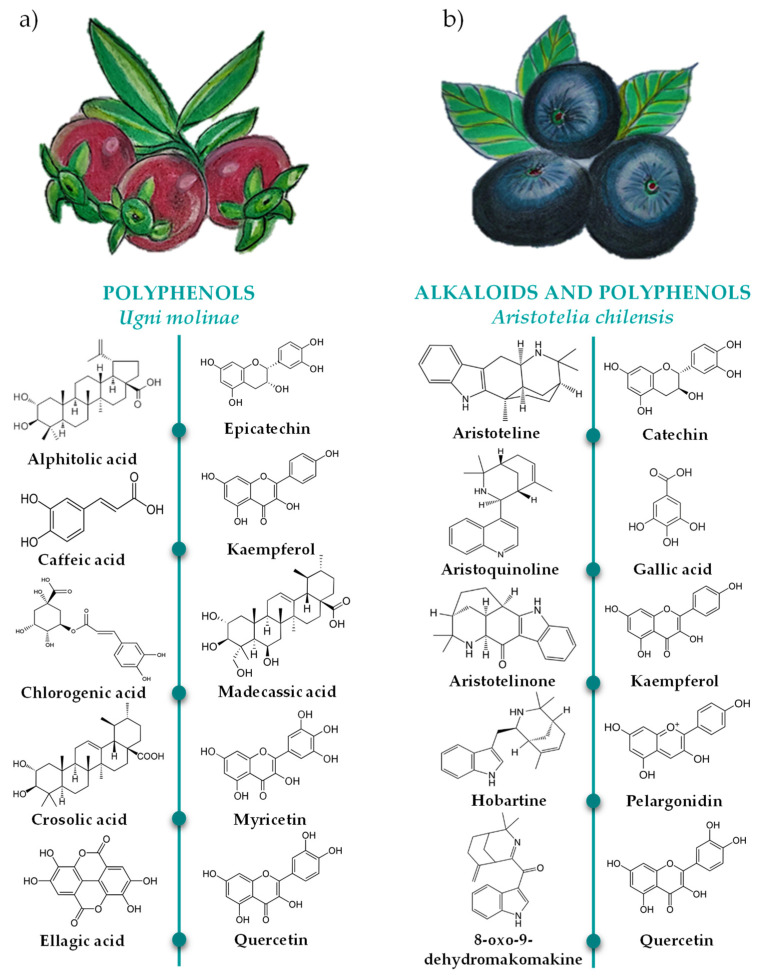
Main bioactive compounds present in the leaves of *Ugni molinae* Turcz (**a**) and *Aristotelia chilensis* (**b**) [73,74,75,76].

**Figure 3 antioxidants-10-01396-f003:**
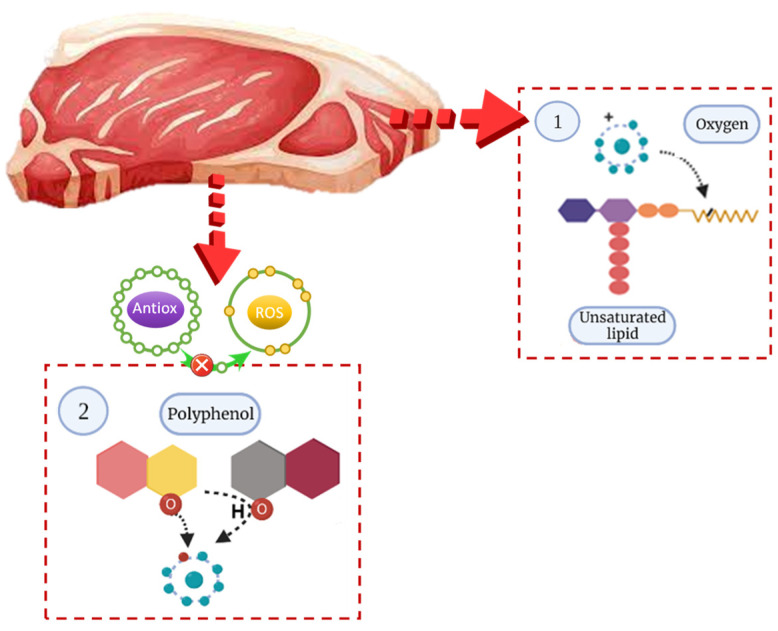
Antioxidant mechanisms of polyphenols. (**1**) Oxygen or reactive oxygen species, deficient in electrons, react with the double bonds of the carbon chains of fatty acids. (**2**) Polyphenol molecule transfers an electron or a proton to the reactive oxygen species, stabilizing it and preventing it from reacting with the fatty acids (created with BioRender.com, 3 July 2021).

**Table 1 antioxidants-10-01396-t001:** Effect of natural extracts obtained from the leaves of different endemic plants on the shelf life of processed meat products.

Plant Leaves	Meat Product	Extract Dose	Storage Conditions	Main Results	Ref.
Black and green tea (*Camellia sinensis* L.)	Uncured pork frankfurters	0.05, 0.10, 0.20, and 0.30%	5 days at 37 °C	Phenolic extracts inhibited the formation of hydroperoxides. Polyphenol extracts did not affect colour, odour, texture, juiciness, flavour, or general acceptability.	[15]
Black currant (*Ribes nigrum* L.), Cherry (*Prunus cerasus* L.)	Pork frankfurters	BCS 1.0 g/100 g CHS 0.5 g/100 g	28 days at 4 °C	Samples treated with natural extracts showed significantly lower counts of *Brochothrix thermosphacta*, LAB, mesophiles, and psychrotrophs. Polyphenols from leaves of *P. cerasus* and *R. nigrum* inhibited MDA formation by 15.29% and 21.48%, respectively.	[17]
Black currant (*Ribes nigrum* L.), Blueberry (*Vaccinium myrtillus* L.), Garden savory (*Satureja hortensis* L.), Sea buckthorn (*Hippophae rhamnoides* L.)	Cooked meatballs	100 and 200 ppm	14 days at 4 °C	Aqueous extracts inhibited lipid oxidation by 13.8–21.8% during 14 days of cold storage.	[18]
Cadillo (*Bidens pilosa*), *Moringa oleifera*	Ground beef	0.5 and 1 g/kg	6 days at 4 °C	The extracts obtained from the leaves of *M. oleifera* and *B. pilosa* showed a polyphenol content of 77.5 and 75.9 mg GAE/g dw, respectively. Both plants displayed inhibitory effects (1 g/kg) against *Staphylococcus aureus*, *Bacillus cereus*, *Enterococcus faecalis*, and *Escherichia coli*, although the higher activity was observed against gram-negative bacteria.	[32]
Cilantro (*Coriandrum sativum*)	Turkey meatballs	200 and 500 ppm	9 days at 4 °C	The addition of *C. sativum* extract at 500 ppm delayed lipid oxidation by 58.33% during 6 days of storage and inhibited the growth of aerobic microorganisms for 9 days of storage.	[27]
Green tea (*Camellia sinensis* L.), Nettle (*Urtica dioica* L.), Olive (*Olea europea* L.)	Frankfurter sausages	500 ppm	45 days at 4 °C	The ethanolic extracts of *C. sinensis*, *U. dioica,* and *O. europea* leaves inhibited lipid oxidation by 40%, 20%, and 26%, respectively.	[14]
Indian nimbus (*Azadirachta indica*)	Raw, chilled beef patties	0.7% (*w*/*w*)	5 days at 4 °C	The hydroalcoholic extract limited colour loss and reduced metmyoglobin formation (36.70%). It prevented MDA formation by 67.30% after 11 days of storage, which was similar to the values found with the synthetic antioxidant (66.34%). Antimicrobial activity was observed against most of the bacterial strains tested, with the best observed activity against *E. coli* strains.	[8]
Mate (*Ilex paraguariensis*)	Fermented Italian-type sausages	0.3–0.4%	60 days at 18 °C	Aqueous extracts added at 0.4% reduced oxidation by 32.87% during 60 days of storage. Sensory characteristics (flavour, texture, and overall acceptability) were not affected by the addition of *I. paraguariensis* extract.	[33]
Oregano (*Origanum vulgare*)	Marinated pork	0.5–2%	10 days at 4 °C	Aqueous and ethanolic extracts of *O. vulgare* added at 2% inhibited lipid oxidation by 63.2% during 10 days of storage. The extracts showed inhibitory power against the growth of Enterobacteria and Enterococci.	[34]
Pitanga (*Eugenia uniflora*)	Pork burgers	250, 500 and 1000 ppm	18 days at 4 °C	Hydroalcoholic extracts showed to inhibit the growth of *Salmonella* spp., *Bacillus cereus*, *Staphylococcus aureus*, and *Pseudomonas aeruginosa*.	[29]
Rosemary (*Salvia Rosmarinus*)	Frozen pork, turkey and chicken patties	0.03% and 0.06%	120 days at −12 °C	The proximate composition of the different formulations did not show significant differences. The concentration of natural antioxidants was sufficient to maintain the oxidative stability of the products during frozen storage.	[16]
	Duck meatballs	1%	15 days at 3 °C	Meatballs with extracts of *S. rosmarinus* inhibited hydroperoxide formation by 80.7%. The counts of psychrotrophic and coliform bacteria were lower in meatballs containing rosemary (≤10 cfu/g) compared to those found in control sample (1.3 × 10^4^ and 8.7 × 10^2^ cfu/g, respectively).	[35]
Salvia (*Salvia leriifolia*)	Burgers	5, 10, 15 and 20 mg/L	45 days at −12 °C	The inclusion of powdered leaves of *S. officinalis* decreased the growth of *S. aureus* and TVC. This effect was significant at 15 and 30 days of storage, respectively.	[36]
Walnut (*Juglans regia* L.)	Pork meat	Extract (5.5%) or powder (0.5%)	15 days at 4 °C	Aqueous extracts reduced MDA formation by 47.5% after 15 days of storage compared to controls, including samples without antioxidants and samples with 0.1% BHT.	[37]

BCS, blackcurrant leaf extract; CHS, sour cherry leaf extract; dw, dry weight; MDA, malonaldehyde; TVC, total viable counts.

## Data Availability

Not applicable.

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
