# Peer review of "Natural Antioxidants from Endemic Leaves in the Elaboration of Processed Meat Products: Current Status"

_antioxidants, 2021, doi:10.3390/antiox10091396_

Round 1
Reviewer 1 Report
The title of the manuscript implicates that it deals with the use of two chilean endemic plants (Aristotelia chilensis and Ugni molinae) in meat processing. In fact, it summarizes the information on the use of other plant antioxidants in meat industry on one hand and the information on antioxidant content / activity in Aristotelia chilensis and Ugni molinae on the other hand. The connection between these two plants antioxidants and meat processing is only speculative, not supported by any data and thus not discussed at all. In my opinion, this is the major deficiency of the review manuscript.
Also the English of the manuscript should be improved. The text is prepared carelessly, for example there are lacking words in some sentences, some expressions are doubled, etc.
Fig. 3 is confusing. The reaction of polyphenols with free radical is not a result of the reaction of molecular oxygen with lipid molecule as it can be deduced from the figure.
Author Response
Reviewer 1
Comments and Suggestions for Authors
The title of the manuscript implicates that it deals with the use of two chilean endemic plants (Aristotelia chilensis and Ugni molinae) in meat processing. In fact, it summarizes the information on the use of other plant antioxidants in meat industry on one hand and the information on antioxidant content / activity in Aristotelia chilensis and Ugni molinae on the other hand. The connection between these two plants antioxidants and meat processing is only speculative, not supported by any data and thus not discussed at all. In my opinion, this is the major deficiency of the review manuscript.
Answer: In this review we want to reflect the importance of the use of antioxidants and natural antimicrobials today to increase the stability and shelf life of processed meat products, especially those that come from the leaves of endemic plants. This is reflected in the title of the manuscript “Natural antioxidants from endemic leaves in the elaboration of processed meat products: current status”. The extracts obtained from the leaves of many of these plants (Bidens pilosa, Camellia sinensis, Coriandrum sativum, Eugenia uniflora, Moringa oleifera, Olea europea, Origanum vulgare, Prunus cerasus, Ribes nigrum, among others) have already been incorporated into meat products, where it has been possible to evaluate their positive effects on safety, oxidative stability, and organoleptic characteristics during the shelf life of these products. In addition to these, we wanted to present two other endemic plants (Aristotelia chilensis and Ugni molinae Turcz) with great potential to be incorporated in processed meat products. The connection between all the information included in this review is the evaluation of the polyphenolic content of the leaves of endemic plants and its potential as antioxidants and antimicrobials in meat products.
Also the English of the manuscript should be improved. The text is prepared carelessly, for example there are lacking words in some sentences, some expressions are doubled, etc.
Answer: Thank you for your comment. The review has been carefully checked by an English speaker.
Fig. 3 is confusing. The reaction of polyphenols with free radical is not a result of the reaction of molecular oxygen with lipid molecule as it can be deduced from the figure.
Answer: We are grateful for your comment that undoubtedly helped us to improve the manuscript. The figure 3 has been improved.
Reviewer 2 Report
Very nice review of the subject with a focus on three plant species for use as sources of chemoprotectants for meat products. I recommend acceptance.
Author Response
Reviewer 2
Comments and Suggestions for Authors
Very nice review of the subject with a focus on three plant species for use as sources of chemoprotectants for meat products. I recommend acceptance.
Answer: Thank you very much for your comment that recognize our work.
Reviewer 3 Report
The Introduction section should include all species mentioned in the manuscript.
Regarding the antimicrobial effects of Bidens pilosa reported in Table 1 (reference #31), authors could also consider to improve the description considering recent findings of antimicrobial effects of different plant materials of this species against Gram+ and Gram-, but also fungi strains. (Angelini P, Matei F, Flores GA, Pellegrino RM, Vuguziga L, Venanzoni R, Tirillini B, Emiliani C, Orlando G, Menghini L, Ferrante C. Metabolomic Profiling, Antioxidant and Antimicrobial Activity of Bidens pilosa. Processes. 2021; 9(6):903. https://doi.org/10.3390/pr9060903).
The quality of figure 2 should be improved.
There are different typos in the names of phytochemicals. Please check ellagic acid, caffeic acid, gallic acid. These typos are present in figure 2.
Lines 524-526: please correct the sentence.
Author Response
Reviewer 3
Comments and Suggestions for Authors
The Introduction section should include all species mentioned in the manuscript.
Answer: Thank you for your comment. The introduction has been improved according to the reviewer's suggestion.
Regarding the antimicrobial effects of Bidens pilosa reported in Table 1 (reference #31), authors could also consider to improve the description considering recent findings of antimicrobial effects of different plant materials of this species against Gram+ and Gram-, but also fungi strains. (Angelini P, Matei F, Flores GA, Pellegrino RM, Vuguziga L, Venanzoni R, Tirillini B, Emiliani C, Orlando G, Menghini L, Ferrante C. Metabolomic Profiling, Antioxidant and Antimicrobial Activity of Bidens pilosa. Processes. 2021; 9(6):903. https://doi.org/10.3390/pr9060903).
Answer: We are grateful for your comment that undoubtedly helped us to improve the manuscript. The discussion of the antimicrobial effects of Bidens Pilosa has been improved including the proposed reference. We would like to mention that the information has not been included in the table since it refers to the effects observed in meat products.
The quality of figure 2 should be improved.
Answer: The figure 2 has been improved.
There are different typos in the names of phytochemicals. Please check ellagic acid, caffeic acid, gallic acid. These typos are present in figure 2.
Answer: These mistakes have been corrected.
Lines 524-526: please correct the sentence.
Answer: Thank you for your comment. The sentence has been modified.